# Utility of Animal Models to Understand Human Alzheimer’s Disease, Using the Mastermind Research Approach to Avoid Unnecessary Further Sacrifices of Animals

**DOI:** 10.3390/ijms21093158

**Published:** 2020-04-30

**Authors:** Tian Qin, Samantha Prins, Geert Jan Groeneveld, Gerard Van Westen, Helga E. de Vries, Yin Cheong Wong, Luc J.M. Bischoff, Elizabeth C.M. de Lange

**Affiliations:** 1Predictive Pharmacology, Division of Systems Biomedicine and Pharmacology, Leiden Academic Centre of Drug Research, Leiden University, 2333 CC Leiden, The Netherlands; t.qin@lacdr.leidenuniv.nl (T.Q.); l.j.m.bischoff@lacdr.leidenuniv.nl (L.J.M.B.); 2Centre for Human Drug Research (CHDR), 2333 CL Leiden, The Netherlands; SPrins@chdr.nl (S.P.); GGroeneveld@chdr.nl (G.J.G.); 3Computational Drug Discovery, Division of Drug Discovery and Safety, Leiden Academic Centre of Drug Research, Leiden University, 2333 CC Leiden, The Netherlands; gerard@lacdr.leidenuniv.nl; 4Neuro-immunology research group, Department of Molecular Cell Biology and Immunology, Amsterdam Neuroscience, Amsterdam UMC, 1081 HZ Amsterdam, The Netherlands; he.devries@amsterdamumc.nl; 5Advanced Modelling and Simulation, UCB Celltech, Slough SL1 3WE, UK; eric.wong@ucb.com

**Keywords:** Alzheimer’s disease, early diagnosis, body fluids, biomarker, animal models

## Abstract

To diagnose and treat early-stage (preclinical) Alzheimer’s disease (AD) patients, we need body-fluid-based biomarkers that reflect the processes that occur in this stage, but current knowledge on associated processes is lacking. As human studies on (possible) onset and early-stage AD would be extremely expensive and time-consuming, we investigate the potential value of animal AD models to help to fill this knowledge gap. We provide a comprehensive overview of processes associated with AD pathogenesis and biomarkers, current knowledge on AD-related biomarkers derived from on human and animal brains and body fluids, comparisons of biomarkers obtained in human AD and frequently used animal AD models, and emerging body-fluid-based biomarkers. In human studies, amyloid beta (Aβ), hyperphosphorylated tau (P-tau), total tau (T-tau), neurogranin, SNAP-25, glial fibrillary acidic protein (GFAP), YKL-40, and especially neurofilament light (NfL) are frequently measured. In animal studies, the emphasis has been mostly on Aβ. Although a direct comparison between human (familial and sporadic) AD and (mostly genetic) animal AD models cannot be made, still, in brain, cerebrospinal fluid (CSF), and blood, a majority of similar trends are observed for human AD stage and animal AD model life stage. This indicates the potential value of animal AD models in understanding of the onset and early stage of AD. Moreover, animal studies can be smartly designed to provide mechanistic information on the interrelationships between the different AD processes in a longitudinal fashion and may also include the combinations of different conditions that may reflect comorbidities in human AD, according to the Mastermind Research approach.

## 1. Introduction

Alzheimer’s disease (AD) is a complex progressive neurodegenerative disorder and is the most common cause of dementia. AD can roughly be divided into two types: familial AD (~5% of total AD patients) and late-onset (LOAD) or sporadic AD (~95% of total AD patients). Familial AD is caused by mutations in either the *APP* gene or in the genes encoding presenilin 1 *(PSEN1)* or presenilin 2 *(PSEN2)*, which are essential components of the γ-secretase complex [1]. These mutations lead to the elevation of total amyloid beta (Aβ), a higher Aβ_1-42_/Aβ_1-40_ ratio, and Aβ aggregation [2]. In sporadic AD, the disturbance of Aβ clearance mechanisms is thought to be the major contribution to Aβ accumulation in the brain, but a (causal) relationship is not fully understood [3,4]. In contrast, it is well-established that an increased frequency of the *ApoE ε4* allele indicates increased risk to develop AD [5,6]. The *ApoE ε4* allele plays an important role in several AD-related processes, such as the oxidative stress response [7], synaptic loss [8], Aβ accumulation [9], and ApoE/LRP1-mediated Aβ clearance [4]. Studies with transgenic *APOE^−/−^* mice showed that these mice develop blood–brain barrier (BBB) breakdown. *APOE4* drives the matrix metalloproteinase 9 (MPP-9)-mediated BBB dysfunction that finally contributes to disturbed influx/efflux of Aβ across the BBB [10].

Different stages in AD progression have been defined [11,12]: the first is the preclinical stage or asymptomatic stage. It occurs between the earliest pathogenic events of AD and the first appearance of specific cognitive changes, which are different from the changes observed in normal ageing. This asymptomatic stage might take many years to develop [13,14]. The second stage is the prodromal stage and is defined by mild cognitive impairment (MCI). In this stage, cognitive changes and amnestic symptoms are present. Importantly, MCI is not selective for AD as not all individuals with MCI develop AD, but individuals with MCI have an increased risk of developing AD or other forms of dementia [15]. In the third and final stage of AD, brain Aβ plaques and neurofibrillary tau tangles (NFTs) may appear on imaging tests of the brain. Individuals at this stage lose control of physical functions and depend on others for care. They sleep more often and are unable to communicate or even recognize their loved ones.

Currently there is no treatment for AD other than some symptomatic treatments that do not slow down or halt AD progression. It is thought that treatment options for AD modification will be more effective during the preclinical stage [11,16,17,18,19]. Postmortem autopsy of the AD brain, which then shows atrophy, neuronal loss, Aβ plaques, and NFTs, is the only certain AD diagnosis [20,21]. During life, clinical evaluation of AD considers cognitive deficits by neuropsychological assessments and measurements of Aβ_1-42_ and total tau (T-tau) in cerebrospinal fluid (CSF) [22,23]. The CSF Aβ_1-42_ level and Aβ_1-42_/Aβ_1-40_ ratio have now been widely accepted as valid indicators of brain accumulation of Aβ [24]. Furthermore, imaging techniques like magnetic resonance imaging (MRI) and positron emission tomography (PET) are used to obtain information on Aβ plaques and the size of the brain and to rule out possible other causes of dementia.

The diagnosis of early AD is currently not yet possible, and there is a great need for information regarding and understanding of the processes that are involved in the onset and early stages of AD. Currently, subjective cognitive decline (SCD) epidemiological data provide evidence that the risk for mild cognitive impairment and dementia is increased in individuals with SCD [25], but we do not yet know what mechanism drives the body toward developing AD. Thus, we have a gap in our understanding of onset and early development of AD.

The problem challenge facing this field of research is that of obtaining more mechanistic information on the time course and interrelationships of the rate and extent of processes that drive the onset and early development of human AD. In humans, there is the possibility for monitoring blood levels of multiple body compounds (potential biomarkers) in cohort. Many such cohort measurements are currently ongoing. Although we might learn a lot from such studies, there are crucial limitations. First, for detecting early changes in body processes that may lead to AD, plasma information is not sufficient, as the levels of body compounds may result in many disturbances not necessarily connected to AD onset. Information on the brain might be provided by what can be detected using imaging techniques. However, imaging techniques are very costly and will not be used in all subjects, let alone in each human subject at each year of follow-up. Thus, human subjects will have developed (significant indicators of) AD before the information of the human subject in relation to AD progression can be obtained. As AD progresses slowly, this will take years at least. In other words, such studies at best would be very expensive and time-consuming. Therefore, we look to additional, alternative approaches to help solve the problem.

Animal models of AD do not really reflect AD in humans. Human AD is familial for only about 5% of cases, while most animal AD models are based on mutations in *APP*, *PSEN1*, and/or *PSEN2* genes. However, the “artificial” AD in animal models of AD might still provide us with information that can be helpful to unravel processes associated with development of AD. This could be helpful in guiding research in humans, focusing on what can be learned from body fluids that can readily be obtained from humans. In animal AD models we can also investigate the influence of combinations of conditions in a well-defined setting [26]. As an example, in mice, it was shown that a combination of ApoE deficiency and high-fat diet, but not these conditions on their own, leads to BBB disruption and neuropathology [27], as would be useful in research on comorbidities in AD.

In this review, we first provide an overview of pathophysiological hypotheses/mechanisms and underlying processes that are thought to play a role in the onset and progression of AD. Then, we summarize published data on the frequently used compounds (biomarkers) in AD research, namely Aβ_1-40_, Aβ_1-42_, hyperphosphorylated tau (P-tau), T-tau, neurogranin, SNAP-25, glial fibrillary acidic protein (GFAP), YKL-40, and neurofilament light (NfL), and their presence/concentrations in brain and body fluids (blood/serum/plasma and CSF) in human AD subjects and controls and in frequently used animal models of AD. Next, we compare the human and animal data to identify similarities. Furthermore, we include emerging new biomarkers that can be measured in body fluids, as these could extend our knowledge on changes in body fluids in AD, to help in identifying composite biomarker panels that indicate the stage of AD and could be used in AD stage diagnosis.

Finally, we discuss the potential value of animal AD models in helping to generate better insights in processes involved in onset and progression of AD, their interactions, and the possibility of designing experiments with well-defined conditions (e.g., to include comorbidities) to understand the influence of these conditions and combinations thereof.

## 2. Pathophysiological Hypotheses and Associated Biomarkers

To effectively identify and validate biomarkers of AD, knowledge about the underlying molecular pathogenesis of AD is critical. A comprehensive overview of all pathological processes is needed to understand the disease. As AD is a complex disease, in which multiple processes are known to play a role, different hypotheses exists focusing on the different processes which (might) play a role in AD [28,29,30]. Overall, these are briefly described in Appendix A, and the timeline is displayed in Figure 1.

Some recent findings seem to provide new hypotheses. These include changes in the functioning of the endocrine pathway and the vagus nerve [38] as well as in the gut-microbiome-derived metabolites [29]. In addition, not so long ago, glucose hypo-metabolism was found as an early pathogenic event in the prodromal phase of AD and was associated with cognitive and functional decline [39], which makes glucose metabolism brain-imaging 2-deoxy-2-fluorine-18-fluoro-d-glucose positron emission tomography (18FDG-PET) a valuable indicator for the diagnosis of neurodegenerative diseases that cause dementia, including AD [40].

Cerebral insulin resistance has been accepted as contributing to the neurodegenerative process in AD by activating oxidative stress, cytokine production, and apoptotic process [41]. It is also the link between sporadic AD and its risk factor of diabetes [42]. Furthermore, age-related decline in the ability of glucose to cross the BBB might lead to the production of Aβ plaques and tau-containing neurofibrillary tangles (neuro-energetic hypothesis) [43]. Moreover, insulin resistance in type 2 diabetes mellitus and obesity is a risk factor for AD, as insulin resistance might contribute to neurodegeneration [44].

Interestingly, type 2 diabetes mellitus animals develop accumulation of AD pathologies like Aβ plaques and tau phosphorylation [45]. Furthermore, insulin has effects on Aβ production and clearance via the MEK-ERK pathway. Aβ, in turn, is found to induce the removal of cell surface insulin receptors [46]. The contribution of mitochondria in the pathogenesis of AD has also been studied (which relates to oxidative stress); however, their exact role and place in the disease cascade is still not fully known [47]. Furthermore, the disruption of neural circuits has also been studied in relation to AD. Again, the question arises as to whether the disruption of neuronal circuits is a cause or consequence in AD pathology; however, impairment of neuronal circuits is already found in early stages of the disease. Interestingly, some aspects of AD-related neuronal dysfunction reported in mice are quite similar to the human situation [48].

Overall, there are multiple mechanisms that possibly contribute to AD, and these have led to the investigation of a broad range of different biomarkers (Table 1) that are related to these different processes, with neuronal degeneration as a final outcome [49,50,51,52,53,54,55,56,57,58,59,60]. As indicated, biomarkers are urgently needed to provide information regarding the pathobiology of AD in order to find a cure and to diagnose AD, preferably in the preclinical stage, with minimal burden for the patient and minimal costs. Biomarkers obtained from body fluids like CSF and blood are therefore needed. Preferably, these markers will provide information in the preclinical stage.

## 3. Most Frequently Studied Brain-, CSF-, and Blood-Derived Biomarkers of AD in Humans and in Most Frequently Used Rat and Mouse AD Models

First, we selected the most related disease processes in AD (as described in Table 1), and within these processes, we selected the biomarkers that currently have been identified with alteration in body fluids such as CSF and blood in human AD patients. Thus, we ended up with a panel of nine biomarkers, namely, Aβ_1-40_, Aβ_1-42_, P-tau, T-tau, neurofilament light (NfL), neurogranin, SNAP-25, GFAP, and YKL-40. It should be noted that some of these compounds are not specific for AD but may be of value by having a role in AD pathology. Second, we summarized the most frequently used rat and mouse models in AD, to compare the information of these biomarkers in humans and in animals. A detailed literature-searching strategy is provided in Appendix A.

Although all these biomarkers are frequently measured in AD patients, Aβ_1-40_ and Aβ_1-42_ were the most frequently studied ones in AD animal models (as there has been much focus on the amyloid cascade hypothesis). We wondered whether the changes in biomarkers in human and animal body fluids would reflect those in the brain and also whether the biomarker changes in animal models would reflect those found in AD patients. Below, we have provided information on human and animal brain, CSF, and plasma as available for each biomarker. The information has shown to be fragmented such that no direct comparison between animals and humans can be made, but rather a comparison of trends could be made as a heatmap (Figure 2). Our detailed findings on actual data are summarized in Appendix A.

### 3.1. Aβ_1-40_ and Aβ_1-42_

Aβ, especially Aβ_1-42_, is the most frequently studied biomarker of AD. Aβ_1-42_ and Aβ_1-40_ are increased in the brains of AD patients compared to healthy volunteers, as well as in brains of “young” versus “old” animals of different AD rat and mouse models.

In human CSF, Aβ_1-42_ levels are also found to be significantly decreased in the preclinical stage of AD [52,57], while Aβ_1-40_ levels in CSF do not seem to change. In Tg2576 transgenic mice, also no change in Aβ_1-40_ CSF level is found between different age groups (3–23 months); however, there is a significantly decrease in Aβ_1-42_ in the CSF in these rats. Furthermore, like in human AD, a decrease of Aβ_1-42_ in CSF has been reported in older mice compared to young mice for 3xTg-AD, APPPS1, and APP23 AD transgenic mouse models [80,81,82]. Following this, a decreased human CSF Aβ_1-42_/Aβ_1-40_ ratio (as widely accepted as a biomarker with better diagnostic accuracy than CSF Aβ_1-42_ alone for the significantly better association with brain Aβ deposition) [83,145,146,147] was found. Similarly, Maia et al., 2015 found that the CSF Aβ_1-42_/Aβ_1-40_ ratio in the APP23, APP24, and APP51 mouse models decreases with increasing age (i.e., AD progression) [82]. However, this is not a perfect comparison, as in the animal studies, the decrease of Aβ_1-42_/Aβ_1-40_ was with increasing age of the transgenic mice, and this is not perfectly comparable with the cross-sectional comparisons in human studies.

Until recently, Aβ peptides measured in human plasma were considered not, or only slightly, correlated with cerebral β-amyloidosis [57,83], and plasma Aβ_1-40_ concentrations were considered to be influenced by production of platelets with Aβ1-40-loaded α-granules [147]. However, Verberk et al., 2018 concluded that the plasma Aβ_1-42_/Aβ_1-40_ ratio has great potential as a prescreening tool to identify AD pathology [148]. Recently, using advanced immune-precipitation coupled with mass spectrometry analytical platforms, Nakamura et.al., 2018 could indeed indicate that the decreased plasma Aβ_1-42_/Aβ_1-40_ ratio shares the trend in CSF and shows high performance when predicting brain amyloid-β burden [149]. This result is validated by using two independent cohorts of datasets, and both datasets contain cognitively normal individuals, individuals with MCI, and individuals with AD dementia. Furthermore, the plasma APP669-711/Aβ_1–42_ ratio may differentiate between individuals (i.e., cognitively normal individuals, those with MCI, and those with AD) with high and low Aβ load in the brain, with a significant correlation between brain Aβ burden (as measured by amyloid-PET) and CSF Aβ_1-42_ level [149]. Interestingly, the plasma levels of both Aβ_1-40_ and Aβ_1-42_ of Tg2576 transgenic mice also significantly decrease with age (comparing different age groups between 3- and 23-month-old mice). This decrease with age was not observed in the non-transgenic control groups, which had about 100 times lower levels of both Aβ_1-40_ and Aβ_1-42_; these levels did not change with age (6–23 months) [84].

The ratio between P-tau and Aβ_1-42_ is also used. The CSF P-tau/Aβ_1-42_ ratio has been shown to be a sensitive predictive diagnostic biomarker of AD and is recommended by the IWG-2 diagnostic criteria for prodromal AD stage [11,12,150]. Moreover, the human plasma P-tau/Aβ_1-42_ ratio, characterized with tau-PET, has recently been indicated to be significantly correlated with cerebral tauopathy [85]. Furthermore, a decrease in plasma Aβ_1-42_ has recently been indicated as a biomarker of prodromal AD progression in patients with amnestic MCI positive subjects (based on Aβ_1-42_/P-tau in the CSF as well as *ApoE ε4* genotype) versus amnesic MCI AD negative subjects [151].

However, compared to the changes found in Aβ_1-40_ and Aβ_1-42_ levels measured in the CSF, we found less consistency in the reported changes in Aβ_1-40_ and Aβ_1-42_ plasma levels in AD patients compared to healthy controls in the different studies. Additionally, no consistency was found in changes in plasma Aβ_1-40_ levels between studies using APP/PS1, 3xTg-AD, or APPPS1 mice, and no consistency was found in plasma Aβ_1-42_ levels between studies using APP/PS1 and 3xTg-AD transgenic mice. However, a decrease of plasma-derived Aβ_1-42_ was reported in the APPPS1 mice, and a decrease was found in both Aβ_1-40_ and Aβ_1-42_ plasma levels in aging Tg2576 mice. No studies were found measuring Aβ_1-40_ and Aβ_1-42_ plasma levels in rat models over time.

### 3.2. T-tau and P-tau

Two different forms of the microtubule-associated protein tau are measured in body fluids: total tau (T-tau) and hyperphosphorylated tau (P-tau). P-tau has decreased capacity to stabilize microtubules. Furthermore, in the brain, P-tau is the primary component of the NFTs [64]. In humans, the deposition of fibrillar tau that aggregates in brain can be accessed with tau PET [152], though with limited level of accuracy for the determination of P-tau versus T-tau. In animal studies, T-tau and P-tau levels in brain tissue homogenate can directly be quantified, and an increase of both P-tau and T-tau levels in brains of older animals compared to young animals was found.

In human CSF samples, P-tau and T-tau were found to be increased in sporadic AD patients compared to healthy controls. In two animal studies, T-tau was measured in CSF. Maia et al., 2013 found an increase in T-tau in APP23 and APPPS1 mice with age [81]. Lecanu et al., 2006 reported an increase of P-tau in CSF in aging FAB rats [86]. Furthermore, P-tau and T-tau were also found to be mostly increased in plasma samples of 3xTg-AD compared to wild-type mice, although this was dependent on the antibody types used [87]. This directly indicates an important point: P-tau has many phosphorylation sites and multiple fragments in body fluids, and the results of measurements depend on the antibody pairs used, which may hamper proper comparison of P-tau data. While the same P-tau form (P-tau181) was measured in most of the human studies [79,85,88,89,90,91,153,154], not all studies specify the P-tau form. In the selected animal studies, the Ser202/Thr205 form was most often used [86,92]; furthermore, less consistency in the use of the same antibody was found in animal studies compared to human studies.

In plasma, P-tau data are variable. Different studies measuring T-tau plasma levels reported an increase [85,93,94] or no significant change in T-tau plasma levels in AD patients compared to healthy controls [94,95,96]. This might be related to the stage of AD or the severity of AD of the AD subpopulation, as a T-tau increase was also not detectable in patients at MCI AD stage [93,94].

### 3.3. NfL

Neurofilament light (NfL) has been shown to be a dynamic cross-disease biomarker for neurodegeneration. Although the increase of NfL in CSF is not specific for AD, it is an important predictor of neurodegeneration in AD as well as cognitive deterioration and structural brain changes over time [155]. In contrast to CSF T-tau and neurogranin alterations, NfL reflects neurodegeneration independent of Aβ pathology [153]. NfL was reported to be increased in human CSF [96,97,98] and plasma [78,96,99,100,101] of AD patients compared to healthy controls. Recently, a study has been conducted, involving patients in a presymptomatic stage of familial AD, in which NfL levels in the CSF (*n* = 187) and serum (*n* = 405) were found to be correlated. Interestingly, the rate of change of serum NfL over time (i.e., log10(serum NfL) per year) could distinguish mutation carriers (i.e., highly penetrant autosomal-dominant mutations in APP, PSEN1, or PSEN2) from nonmutation carriers, almost a decade earlier than cross-sectional absolute NfL levels [156]. These promising results indicate the promise of that blood-based NfL as a valuable biomarker for AD diagnosis in the preclinical stage.

NfL was detected in two transgenic mice models of AD by Bacioglu et al., 2016 [102]. In APPPS1 mice, CSF NfL increased with age: from 3 months to 12 months and to 18 months. Plasma NfL increased accordingly. In the tau-overexpressed P301S-tau mice, CSF NfL increased with age from 2–4 months to 6–8 months and to 10–12 months until 14–16 months, as did plasma NfL [102].

### 3.4. Neurogranin

Neurogranin is a neural-enriched dendritic protein involved in long-term potentiation of synapses, particularly in the hippocampus and basal forebrain. CSF neurogranin is used as a biomarker for synapse loss and synaptic dysfunction in neurodegenerative diseases, including AD. While Kvartsberg et al., 2019 reported a decrease of neurogranin in the brain of AD patients [103], several other studies have reported an increase of neurogranin in CSF of AD patients [79,88,95,104,105,106,157]. CSF neurogranin levels did not significantly differ between AD patients and patients with Lewy body dementia (LBD) [158]. In contrast, Mavroudis et al., 2019 found that CSF levels of neurogranin were significantly higher in AD patients compared with cognitively normal participants, as well as between AD patients and patients with mild cognitive impairment (MCI), So, AD patients have higher neurogranin levels compared to MCI , and MCI has higher neurogranin levels compared to controls, which indicates that neurogranin levels might be used to differentiate MCI patients from AD patients [158]. In blood, no significant change in neurogranin concentration has been detected in AD compared to controls [95,105].

No studies have reported on measuring neurogranin in the brain, CSF, or blood in the selected animal models in a longitudinal fashion. One study using CamKII-TetOp25 mice (inducible transgenic mice overexpressing p25, or Cdk5, which is required for normal development of the mammalian brain) reported a significant increase of neurogranin in the CSF after 3 weeks of p25 induction [159].

### 3.5. SNAP-25

Massive synapse loss is another critical pathological process that occurs in the AD brain and correlates with cognitive decline [160]. Synaptic dysfunction can occur and eventually progresses into massive synaptic loss [161]. This process can be indicated by the decline of synaptic protein levels. Various synaptic proteins, such as SNAP-25, synaptophysin, rab 3A (presynaptic protein), PSD-95, synaptopodin (postsynaptic protein), synapsin 1, and chromogranin B (synaptic vesicle proteins), have been reported to be significantly reduced in the brains of patients with AD [91,107,162,163,164]. Neuronal death alone is not believed to be sufficient to explain the magnitude of synapse loss, suggesting that synapses are selectively damaged or degenerated prior to brain cell death [165]. Reduced gene expression patterns of genes related to synaptic vesicle trafficking have been found, which indicate that synapses in AD may not function effectively, even prior to visible structural alteration of neurons [166]. In contrast to the decrease of SNAP-25 found in human brains, SNAP-25 is found to be increased in the CSF of AD patients compared to age-matched controls [91,108,109,110]. No studies were found measuring the difference of SNAP-25 in the blood of AD patients compared to healthy controls.

Additionally, no studies were found measuring SNAP-25 in brain, CSF, or blood samples in the selected animal models. Reports about synaptic loss in transgenic mice models of AD are mainly based on histological studies. For instance, dendritic spine loss was reported in 8-month-old PDAPP mice and 4.5-month-old Tg2576 mice [167]. Early synaptic loss was identified in hippocampus of 3–4-month-old J20 mice bearing APPswe and APPind mutations. In the brain of these mice, the levels of synaptophysin, PSD95, synaptotagmin, and homer significantly decreased, preceding the deposition of senile plaques [168]. Dendritic spine loss around Aβ plaques began approximately at 3 months of age in APP/PS1 mice [169]. Age-dependent loss of synaptophysin, synaptotagmin, PSD-95, and homer immunoreactivity was reported in the hippocampus of 4-month-old APP/PS1 mice [168]. Although the immunoactivity of certain synaptic proteins has been used to evaluate synaptic loss in these histological studies, the corresponding body-fluid-based changes of these proteins have not been measured.

### 3.6. GFAP

Gliosis is a nonspecific phenomenon that occurs in response to injuries to the brain and involves the activation and proliferation of glial cells. In AD, gliosis is marked by an increase in activated microglia and reactive astrocytes near the sites of Aβ plaques [58]. It is widely accepted that the interaction of microglia with fibrillary Aβ leads to microglia and astrocyte activation, which results in the production of chemokines, neurotoxic cytokines, and reactive oxygen and nitrogen species that are deleterious to the brain [170]. Moreover, it has been shown that reactive astrocytes play an additional role in AD by their contribution to an overall amyloid burden in the brain, given the wide expression of APP, BACE1, and γ-secretase in astrocytes [58]. Glial fibrillary acidic protein (GFAP) is an established indicator for astrocyte activation. GFAP is found to be increased in postmortem human brain tissue samples of AD patients [111], but no significant change is found in postmortem cerebellar brain tissue compared to age-matched controls [112]. An increase in GFAP with age was found in brains from APP/PS1 [113], 3xTg-AD [114], and APPPS1 [115] transgenic mouse models. The same was reported for the McGill-R-Thy1-APP [116] and TgF344-AD [117] transgenic rat model, as well as the FAB rat [86].

Furthermore, GFAP is increased in the CSF of AD patients [98,118]. GFAP levels in serum have also been reported to be increased in serum from AD patients compared to non-neurodegenerative controls, and the increase of serum GFAP correlated with the Mini-Mental State Examination score. Moreover, serum GFAP might be used to discriminate between AD and behavioral variant of frontotemporal dementia [119].

Body-fluid-based markers for gliosis in AD mouse and rat models are not reported.

### 3.7. YKL-40

YKL-40, also called chitinase-3-like 1 (CHI3L1), is a glycoprotein expressed by different cells (such as astrocytes and macrophages) and, although its function is not yet completely understood, is linked to inflammation [120,171]. YKL-40 was found to be increased in the brains of AD patients compared to healthy controls [120]. Furthermore, increased levels of YKL-40 are found in CSF [50,55,89,98,120] and in plasma [50,121] of AD patients compared to healthy controls. Interestingly, Wenström et al., 2015 found increased levels of YKL40 in the CSF of AD patients, compared to the nondemented control group, but not in patients with Lewy body disease or Parkinson’s disease [55]. Moreover, the increase of CSF YKL-40 has also been observed in preclinical (based on clinical dementia rating (CDR) or Mini-Mental State Examination (MMSE) score) AD patients and AD subjects with MCI [50,89]. No studies were found measuring YKL-40 in brain, CSF, or blood samples in the selected animal models.

Overall, for animal studies, researchers tend to seek for “direct answers” of the AD-like pathology in the brain; however, we believe that there is an underestimation of the value in also including body-fluid-based biomarkers of the animals to understand how AD pathology in brain is reflected in body fluids.

## 4. Emerging Techniques and Body-Fluid-Based Biomarkers

In the near future, emerging biomarkers, measured in different body fluids, might provide additional valuable information on processes occurring in the early onset and progression of AD. These include, but are not limited to, extracellular vesicles (EVs); microRNAs; and proteomic-, metabolomic-, and lipidomic-based body fluid biomarkers.

### 4.1. MicroRNAs as Body-Fluid-Derived Biomarkers for AD

MicroRNAs (miRNAs) are a group of biomarkers that can be found in different body fluids, like CSF, blood, and saliva. These small (about 20 nucleotides), noncoding RNAs play a role in many different biological processes. Importantly, miRNAs are known to be conserved across species. Accumulating evidence suggests that alterations in the miRNA networks could contribute to the pathology of AD, or can at least be used as an early indication of the development of AD. Results from recent studies in humans suggest that a number of specific miRNAs are differently expressed in disease conditions, of which some are thought to be involved in the regulation of key genes in AD, including APP and BACE-1.

A subset of miRNAs seems to be specifically altered in the AD brain, including miR-29, miR-15, miR-107, miR-181, miR-146, miR-9, miR-101, miR-106, miR-125b, and miR-132. All were independently validated in two or more studies [60,172]. Furthermore, several large-scale genome-wide profiling studies have been performed, demonstrating that, beside the brain, miRNA levels in blood and CSF are also differently expressed in AD patients, compared to age-matched healthy volunteers [173]. Several miRNAs are indicated to be putatively proinflammatory, including miR-9, miR-125b, miR-146a, and miR-155. The expression levels of these miRNAs are increased in both postmortem brain extracellular fluid (brain ECF) and the CSF of postmortem AD patients [173]. The levels of miRNAs, including miR-137, miR-9, miR-29a, and miR-29b, were found to be significantly reduced in plasma of probable AD patients with a Mini-Mental State Examination score of 23–28 [174].

Recent studies on miRNA expression changes in AD patients suggest the potential of body-fluid-based miRNAs to assist the early diagnosis of AD. However, many of these studies were cross-sectional, with one measurement per patient only, limiting the usage of miRNAs as biomarkers when assessing AD progression. Longitudinal observations of miRNA alterations in AD animal models might provide complementary information of pathology-associated differences in miRNA levels during the progression of AD, particularly in an early stage. Genome-wide analysis of the brain miRNA signature in an APP/PS1dE9 mouse model was investigated by Luo et al., 2014 [175]. In this study, nine miRNAs, namely miR-99b-5p, miR-7b-5p, miR-7a-5p, miR-501-3p, miR-434-3p, miR-409-5p, miR-331-3p, miR-138-5p, and miR-100-5p, showed consistent changes at 2, 4, 6 and 12 months of age in the APPswe/PS1dE9 mouse model. Another study showed 37 miRNAs that are consistently changed in the cerebral cortex of APP/PS1dE9 mice, among which 17 miRNAs are downregulated. These include miR-20a, miR-29a, miR-125b, miR-128a, and miR-106b [176], which are linked to AD based on information of human AD studies. miR-106b is increased in the cerebral cortex of 3- and 6-month-old APP/PS1dE9 mice but decreased in 9-month-old mice (although still remaining slightly higher compared to the level of miR-106b in 3-month-old mice) [177]. This indicates that miRNA expression patterns may change over time.

Furthermore, the plasma miRNA profile was investigated at different timepoints during the AD-like pathology progression in 3xTg-AD mice [178]. Plasma samples were obtained from 2–3-month-old and 14–15-month-old 3xTg-AD and wild-type (WT) mice. No significant differences in miRNA levels were detected between WT and transgenic mice at the young (2–3 months) age, while age-related significant changes in miRNAs were observed in both WT and transgenic mice, with some of these changes being specific for 3xTg-AD mice. Nineteen miRNAs show similar change over time of both WT and transgenic mice. These include family members of let-7, miR-30, and the miR-17-92 cluster and its paralogs. A group of miRNAs, including miR-132, miR-138, miR-146a, miR-146b, miR-22, miR-24, miR-29a, miR-29c, and miR-34a, show significant changes in plasma levels only in the transgenic group. These age-dependent changes are of interest as they could consequently derive from AD pathology progression in this mouse line. The plasma miRNA profile has also been studied over time in the APP/PS1dE9 mice model. At 4 months, when these mice are in the prepathological stage of AD, a significant decrease in expression of miR-200b-3p, miR-139-5p, and miR-27b-3p was observed, together with a significant increase of miR-205-3p and miR-320-3p expression [179]. At 8 months, when amyloidosis is apparent in these mice, the expression of a different set of miRNAs is altered, with an increase in 4 miRNAs (miR-140-3p, miR-486-3p, miR-339-5p and miR-744-5p) and a decrease in miR-143-3p and miR-34a-5p. At 15 months, expression of miR-339-5p and miR-140-3p remained significantly increased, suggesting a sustained increase in expression of these two miRNAs over time.

### 4.2. Proteomic Body-Fluid-Based Biomarkers

Proteomics is a multidisciplinary, technology-driven science that focuses on the analysis of proteomes, i.e., the proteins of a biological system, their structures, interactions, post-translational modifications, and, in particular, the changes in their levels and their modifications as the result of specific diseases or external factors [180].

Although untargeted proteomic analysis can provide us with unbiased body fluid protein panels that have potential diagnostic value in AD, issues around quantification and reproducibility should be considered. Thus, an alternative approach in the proteomic study was advocated where the subjects involved in the studies are grouped to a continuous variable such as brain atrophy, rate of cognitive decline, Aβ burden, and CSF biomarker level (so-called “endophenotype discovery”) [181]. Based on this, Shi et al., 2018 summarized the “endophenotype discovery” for plasma proteomic biomarkers in AD and eight most replicated protein biomarkers were selected [181]. However, it should be noted that for the discovery of robust (i.e., reliable and quantifiable) proteomic biomarkers for AD, the comparability of data from multiple large studies across heterogeneous populations is needed; this remains challenging, as has been addressed by Carlyle et al., 2018 [182].

In case of AD, several CSF- and blood-based proteomics analyses have been conducted by comparing the proteome profiles of these body fluids of AD patients to the control groups to collect information about gene products involved in AD, i.e., alterations in protein levels and post-translational modifications. Several proteomic targets have been discovered that displayed significant alteration in the CSF of AD subjects compared to the control group [183,184,185]. In these studies, a certain consistency was observed in the alteration of apolipoproteins in the CSF of AD patients, suggesting that a pronounced reduction of pro-apolipoproteins might be a potential CSF-based biomarker of AD. In blood, however, only few putative blood-based protein biomarkers could be replicated in independent studies. This was concluded by a large-scale replication check for 94 of the 163 candidate biomarkers from 21 published studies in plasma samples from 677 subjects [186].

In AD animal models, proteomic biomarker profiles from the brain hippocampus homogenate of ADLPAPT mice [187], APP/PS1 mice, and *ApoE4* knock-in mice models [188] have been identified with age-dependent alterations; many of the differentially expressed proteins were identical at presymptomatic stage of the mice, earlier than the formation of Aβ plaque. However, body-fluid-based proteomic biomarkers in AD animal models are still missing. It would be interesting to also investigate these markers in CSF and plasma of the animal AD models to relate these body-fluid-based to brain-tissue-based proteomic markers and the pathological changes of the brain. Furthermore, these biomarkers can be traced throughout the lifespan of animals as well as the disease progression, and this could provide insights in refinement of biomarkers in human body fluids.

### 4.3. Metabolomic and Lipidomic Body-Fluid-Based Biomarkers

Metabolomics is one of the latest systems biology approaches where multiple platforms are utilized to measure levels of small-molecule metabolites in biological samples. Metabolic signatures are unique to an individual wherein perturbations in metabolite levels may inform on the disease state and underlying mechanisms of the disorder [189]. Given their close association with the host’s phenotypes, the profile of metabolites demonstrates the current physiological state of a cell and is the end result of the upstream biological information that flows from genome over to transcriptome and proteome to metabolome [190].

In 2018, Hurtado et al. reported the state of the art in AD-related metabolomic biomarker evidence based on studies on metabolomics and lipidomics in AD [191]; in 2019, several novel targets were reported as potential body-fluid-based biomarkers in AD, where it was found that kynurenine pathway metabolites and primary fatty amides showed great significance in their alterations in AD subjects compared to the controls [192,193]. Multiplatform metabolomics has emerged as an essential tool for the identification of potential AD biomarkers in different human body fluids, and many potential biomarkers have been discovered in the last decade. However, only a few of these have been validated. Moreover, fellow researchers in this area expressed concerns about the consistency of the proteomics, metabolomics, and lipidomics studies and called for interlaboratory validations [190]. So, while this is an emerging field of high interest, it suffers from interlaboratory differences and reproducibility issues. However, different laboratories have extensively validated their metabolic and lipidomic platforms, and series of studies all using the same platform will definitely provide relevant data on changes in AD onset and progression.

### 4.4. Extracellular-Vesicle-Based Biomarkers

EVs include, from small to large size, exosomes, microvesicles, and apoptotic bodies; are secreted from different cells in the body; and contain unique molecular information regarding their cell of origin. They are released into the extracellular environment and are known to play a role in intercellular communication between cells in close proximity, as well as between distant cells. This, together with the fact that EVs have been found in many different body fluids including blood, urine, and CSF, has raised interest in the use of EVs as a source for the discovery of novel biomarkers. It should be noted that the methodologies to isolate EVs, the techniques used for quantification of EVs, and the techniques to quantify their characteristics and content are all crucial for the data obtained, and good comparisons of data currently published are not yet possible [194] (unpublished data of our group). So, what is described below should be interpreted given these drawbacks

Several studies indicate that AD patients could be distinguished from cognitive normal controls based on the (synaptic) protein cargo from neural-derived EVs extracted from plasma. These biomarkers would reflect AD pathology up to 10 years before the clinical onset of AD [195]. Different types of AD biomarkers were investigated by Goetzl et al., showing that Aβ and tau proteins increased along with AD progression [195], while synaptic markers significantly declined [196,197]. In these studies, the EV-based lysosomal proteins, brain insulin resistance factor, and cellular survival factors appeared to be useful in distinguishing between control and AD progression in multiple stages of the disease [198,199]. However, the work of the Goetzl group has not yet been reproduced by others, while their L1CAM work was done in what they claimed as early stages of AD (i.e., not specified/defined), all of which means that the value of their work remains to be seen.

In addition, EVs originating from the Central Nervous System (CNS) were found in plasma, containing the AD-related biomarkers Aβ_1-42_, P-tau, and T-tau. These EVs were found to identify patients converting from MCI to dementia [200]. Moreover, Gui et al., 2015 reported that the miRNA profile in CSF-derived EVs was altered in AD. The mRNA transcripts of APP, Tau, NfL, DJ-1/PARK7, fractalkine and neurosin were altered, and long noncoding RNAs (RP11-462G22.1 and PCA3) were also found to be differentially expressed in CSF-derived EVs [201]. Overall, these studies demonstrate the potential of EV-based biomarkers in the early stages of AD.

Still, only a few studies investigated EV-based biomarkers in AD transgenic mouse and rat models. One study, by Eitan et al., 2016, reported higher levels of Aβ_1-42_ and Aβ_1-40_ in plasma-derived EVs of six APP/PS1 mice and five 3xTgAD mice compared to nine age-matched WT control mice. The absolute levels of Aβ_1-42_ and Aβ_1-40_ in plasma-derived EVs in these transgenic mice were lower compared to the levels of these markers found in EV-depleted plasma, while the ratio of Aβ_1-42_/Aβ_1-40_ was significantly higher in EVs [202]. This indicates that EV-based biomarker alterations can be different from body-fluid-based alterations and that plasma-derived EVs might provide biomarkers with higher sensitivity than whole-plasma-derived biomarkers.

## 5. Discussion and Conclusions

The number of currently existing and emerging pathophysiological hypothesis, mechanisms, theories, and processes related to AD is high and is still increasing. This indicates that AD is a very complex and multifactorial disease. An important problem is that we do not have a cure or treatment for AD. Another problem is that the disease cannot be diagnosed in an early stage. Currently, we lack information and understanding of processes in the onset and early stage of the disease. As such, we lack an early diagnosis and treatment option in the early phase of AD. This highlights the need to find adequate, preferably body-fluid-based biomarkers of AD. Currently, the biomarkers that are mostly measured in human studies are Aβ, P-tau, T-tau, neurogranin, SNAP-25, GFAP, YKL-40, and especially NfL. Additionally, there is a high volume of animal research, in which the emphasis has mostly been on Aβ.

For early diagnosis and treatment of AD, we first need to solve the problem of the gap in the knowledge and understanding of the onset and early phase of AD. miRNAs and EVs, together with proteomic, metabolomic, and lipidomic body-fluid-based biomarkers are emerging as (early) biomarkers of AD as well as other diseases. miRNAs are conserved across species, which makes it easier to extrapolate findings between humans and animal models of AD. However, as a drawback, it is hard to assess whether a change in microRNA expression level is a result or a cause of AD. Moreover, a single microRNA can target multiple genes, and one gene can be targeted by different microRNAs. Furthermore, while omic technics show great potential in the discovery of new biomarkers in AD, a drawback here is the current lack of consistency and reproducibility, which indicates that omic markers in AD need to be further validated. Then, the use of EVs combined with the measurement of EV-associated and non-EV-associated miRNAs together with techniques like metabolomics and lipidomics show great promise for the detection of novel biomarkers in body fluids and for the collection of new information to increase our understanding of the pathogenesis of AD. Here, the drawback is in the multiple methodologies in isolating and characterizing EVs, which influence the data obtained and thereby make comparisons between the data difficult.

For the biomarkers that have been investigated in brain, CSF, and blood, the changes in disease stage are rather different (Figure 2), and so they are not directly related. This indicates that body fluids might not directly provide mechanistic information on the disease stage, and in humans it might be difficult to study the interrelationships and time-dependencies of the biomarkers. Therefore, we need different approaches for more mechanistic understanding of early AD and its progression.

We believe that the problem of the gap in knowledge and understanding of the onset and early phase of AD cannot be solved by human studies alone. This is for the simple reasons that it is too costly and too time-intensive to measure many compounds in human samples, let alone imaging studies, in well-controlled and longitudinal studies in the ageing human population, in which a small percentage will actually develop AD. Thus, there is the need for alternative approaches to obtain a useful understanding on the relation between the changes in (body fluid) biomarkers and (early) AD stage. In our view, animal studies could be helpful.

An ideal animal model that exhibits all features of human AD does not exist. Current animal AD models are at best to be regarded as reductionist tools, as the majority of the animal models represent familial AD while most AD patients have sporadic AD (although a few non-transgenic rat models of AD have been developed [117]). However, animal models of AD may still be of value to gain understanding on the pathological processes involved in AD if enough similarities exist between animal AD models and human AD. To that end, an overview is given on associated and most frequently measured biomarkers in brain, CSF, and blood of human AD and animal models of AD. Where possible, the trend in changes of these biomarkers is assessed and summarized in a heat map (Figure 2).

Although a direct comparison between human (familial and sporadic) AD and (mostly genetic) animal AD models cannot be made, a majority of similar trends are observed in brain, CSF, and blood for human AD stage and animal AD model life stage, assuming that a later stage in life of the AD animal represents a later stage in AD. Despite the current limitation of exact knowledge on AD stage in humans and in animals, we see many similarities. This makes us believe that animal models of AD have a good potential to provide information that can be useful for also better understanding the (early) processes in AD.

While we all strive for the reduction/replacement in the use of animals, we should realize two things. (1) The problem of AD is too big not to make all efforts possible to diagnose AD in an early stage, where the disease might be halted or even reversed. (2) AD is too complex to be understood from single-biomarker and single-timepoint measurements. Therefore, we must study this complex disease in a very systematic research manner including multiple biomarker measurements at multiple sites in the body (fluids and tissue) in a longitudinal fashion under well-defined conditions, applying advanced mathematical modeling (according to the Mastermind Research approach [26]) to unravel the processes and their interactions (Figure 3). Ideally, composite biomarker panels will reflect all processes that occur in AD in a stage-dependent manner. However, it should be noted that due to the small sample size of brain ECF, CSF, and blood that can be obtained from rats and especially mice, sampling as well as detecting compounds and EVs in these body fluids is challenging.

Recently, we have shown that the strategic use of animals and the collection of smart data have led to a mathematical model that can adequately predict drug distribution into multiple physiologically relevant compartments of the CNS, not only in animals, but also in humans. It should be noticed that CNS drug distribution is also the result of multiple processes and their interdependencies. The CNS drug distribution model has been developed using systematic research in experimental animals by varying conditions, measuring drug concentrations at multiple locations in the CNS, and making differences between drug and body properties explicit, such that the body properties of the rat could be replaced by human body properties. Our CNS drug distribution model can now replace the use of animals and directly predict CNS drug distribution in humans on the basis of plasma pharmacokinetics and drug properties [203,204]. This indicates a much better and efficient use of animals.

Thus, for AD, longitudinal early-life-studies can be performed in both transgenic and non-transgenic animal AD models and their control littermates, on a much shorter time scale than in humans, while measurements can also be taken at multiple timepoints at multiple body sites (including body fluids) and finally also in the brain, to relate to currently known brain markers of AD. This anticipated approach is depicted in Figure 4.

Altogether, in our view, strategic and well-controlled animal studies are needed to fill the crucial knowledge gap, especially on the processes involved in the onset and early stage of AD. Thus, as much as possible in individual animals, multiple (putative) biomarkers should be measured at multiple body sites, including body fluids, to understand their interdependencies and time-dependencies in AD onset and progression (according to the Mastermind Research approach). By this approach, systematic research can also be performed on combinations of conditions, such as comorbidities.

Once we have such understanding, we will have a good basis for defining (multiple) targets to be modified by therapeutic approaches in order to halt the disease or even be able to reverse the disease in its early stage into a healthy condition again.

## Figures and Tables

**Figure 1 ijms-21-03158-f001:**
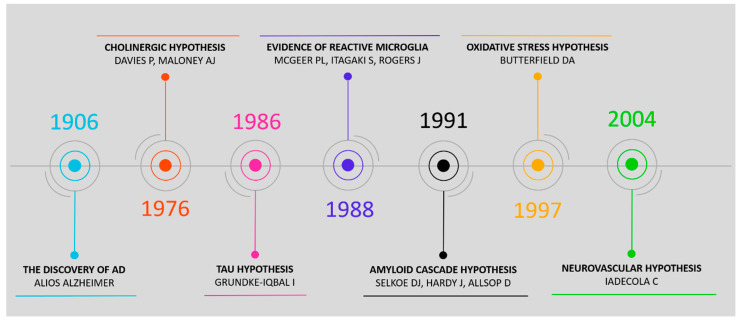
Timeline showing the pathophysiological hypotheses of Alzheimer’s disease: the cholinergic hypothesis [31], tau hypothesis [32], evidence of reactive microglia [33], amyloid cascade hypothesis [34,35], oxidative stress hypothesis [36], and neurovascular hypothesis [37].

**Figure 2 ijms-21-03158-f002:**
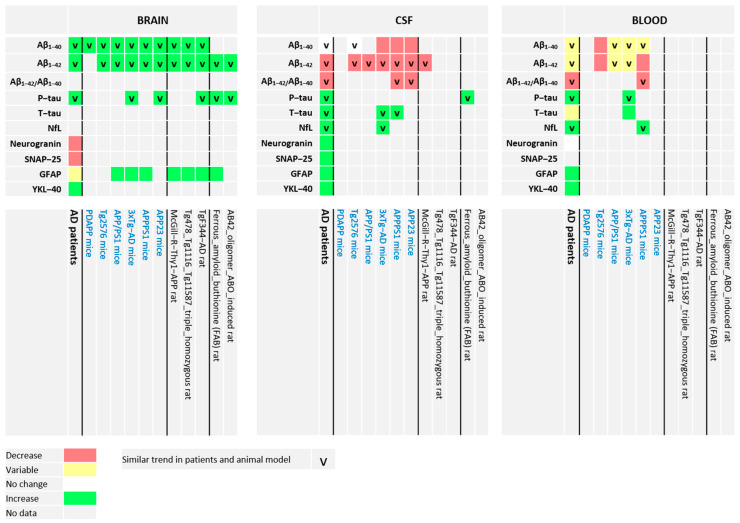
Heat map of the changes of the selected biomarkers measured in the brain, CSF, and blood in humans and different animal models of AD. In humans, the change in biomarker level is compared between AD patients and healthy volunteers. In AD animal models, the change in biomarker level is compared between “old” and “young” AD animals (different age groups) and therefore the change during life in which AD features are assumed to progress. Literature used: [50,52,55,78,79,80,81,82,83,84,85,86,87,88,89,90,91,92,93,94,95,96,97,98,99,100,101,102,103,104,105,106,107,108,109,110,111,112,113,114,115,116,117,118,119,120,121,122,123,124,125,126,127,128,129,130,131,132,133,134,135,136,137,138,139,140,141,142,143,144] (for details on values of individual studies please see also Appendix A).

**Figure 3 ijms-21-03158-f003:**
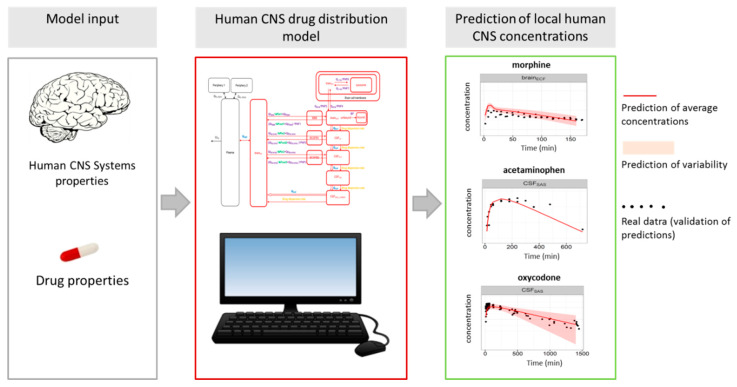
The recently developed human CNS drug distribution model is an example of application of the Master Research Approach [26]. The model is developed on the basis of animal research, and now CNS drug distribution in humans can be predicted without the need of experimental animals [203,204].

**Figure 4 ijms-21-03158-f004:**
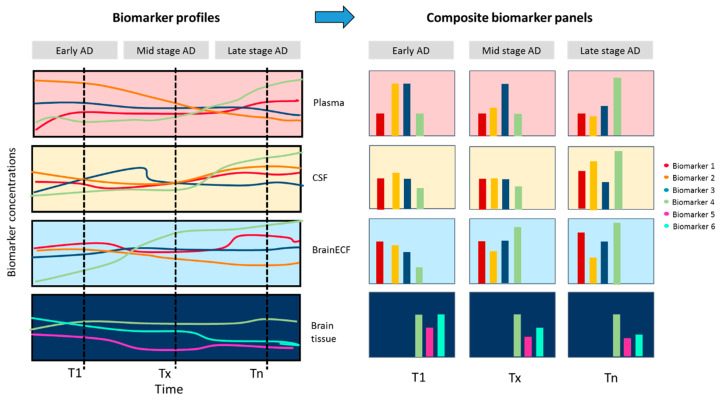
Anticipated approach to study and understand the processes in AD progression. Longitudinal, multiple-biomarker, multiple-body-site measurements in AD animals (and their control littermates—not shown here) should be able to reveal processes and their interdependencies in AD and in normal ageing as stage (T1, Tx, Tn)-dependent “composite biomarker panels”, leading to insights that are AD-specific to be targeted as therapy.

**Table 1 ijms-21-03158-t001:** Alzheimer’s disease (AD)-related processes and the corresponding potential body-fluid-based biomarkers. It should be noted that some of these compounds are not specific for AD but may be of value by having a role in AD pathology.

Process	Remarks	Related (Potential) Body-Fluid-Based Biomarkers	Reference
Decreased cholinergic transmission	Not a definitive causation of the disease, but merely a consequence	Acetyltransferase (ChAT)	[54]
		Acetylcholinesterase (AChE)	[54]
		SNAP-25	[61]
Dysfunction in phosphorylation process of tau protein resulting in hyperphosphorylation of the molecule	Secondary pathogenic event that subsequently causes neurodegeneration in AD	Total tau (T-tau) Hyperphosphorylated tau (P-tau)	[51,60,62,63,64]
Reactive gliosis and neuroinflammation	Reactive microglia and astrocytes surround amyloid plaques and secrete proinflammatory cytokines, which are an early, prime movers in AD evolution	Glial fibrillary acidic protein (GFAP)	[49,58]
		S-100B	[49,58]
		YKL-40	[50,55]
Inequality between production and clearance leads to amyloid β (Aβ) accumulation in brain	The triggering event and the most important factor with highest acceptance but still not exclusively the cause of the disease	Aβ_1-42_ Aβ_1-40_	[24,52]
Characteristic presence of oxidative stress in AD brains	Reactive oxygen species (ROS) and neuronal apoptosis are involved not only in AD but also other neurodegenerative diseases. Below, we propose the oxidative stress pathways specific to AD and involved kinases as potential biomarkers for these processes	-	-
	N-Methyl-D-aspartate receptor (NMDR)-mediated oxidative stress inducing abnormal hyperphosphorylation of tau	Mitogen-activated protein kinase (MAPK) and extracellular receptor kinase (ERK)	[65,66]
		Calmodulin-dependent protein kinase (CaMKII)	[67,68]
	Aβ activates GSK-3β, which induces oxidative stress, resulting in hyperphosphorylation of tau, NFT formation, neuronal death, and synaptic loss	Glycogen synthase-3β (GSK-3β)	[69,70]
	NMDR-mediated oxidative stress leads to activation and phosphorylation of CREB	cAMP response element-binding protein (CREB)	[71,72,73]
	Calcineurin activation leads to release of intracellular Ca^2+^ and reduced NMDR function. Aβ reduces NMDR function whichimpairs LTP through enhanced calcineurin activity	Calcineurin	[74,75]
Cerebrovascular dysfunction, alterations in cerebral blood flow, and impairment of low-density lipoprotein receptor-related protein-1 (LRP-1).	Morphological alterations in cerebral capillaries and increased use of CBF and glucose utilization have been reported in AD patients	LRP-1	[76,77]
Neurodegeneration	Endpoint of different processes	Neurofilament light (NfL)	[78]
		Neurogranin	[79]
		SNAP-25	[61]

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
