# Peer review of "Utility of Animal Models to Understand Human Alzheimer’s Disease, Using the Mastermind Research Approach to Avoid Unnecessary Further Sacrifices of Animals"

_ijms, 2020, doi:10.3390/ijms21093158_

Round 1

Reviewer 1 Report

The authors make a comparative review (models and clinical data) of biomarkers of Alzheimer´s disease (AD) with an introduction on different hypotheses of the disease. The narrative is good, the rationale convincing and timely, but there are several points that need to be addressed. In particular, the topic has an extensive literature, but the authors have favored to include specific publications over many others. They should clarify why they chose to omit so many and the way they selected them (which is almost inevitable, given the enormous amount of data). They should also clearly state criteria to select specific biomarkers.

At any rate, the main idea: more data should be obtained in appropriate animal models; is quite interesting.

Specific comments:

Hypotheses of AD: should be nuanced. Thus, dysfunctional microglia is not hypothesized as the origin of AD; maybe pathogenic in the cascade, maybe not. This is still not clear. Oxidative stress may be contributing to the pathological cascade. It is no longer considered a mainstream cause of the disease. There are other hypotheses that are not mentioned, such as disturbed insulin function, dysregulated circuitry ….etc. If the authors considered them not worthy to be included, they should specify their selection criteria.

Table 1: Useful and interesting, but re-arrange it, more precision is needed. Many of the putative biomarkers are 1) arbitrarily assigned to a given hypothesis, or 2) arbitrarily added. I will elaborate: 1) For instance, Neurofilament, included in the “cholinergic” (outdated, by the way) hypothesis. Neurofilament is been considered as a marker of neuronal degeneration not only in AD but in neurodegenerative diseases in general (as commented by the authors also). Those added in the “Oxidative” hypothesis (calcineurin,CREB…) could be equally well included in other hypotheses, as they are involved in many biological pathways. 2) There are many more potential AD biomarkers reported. Maybe too many, making a lot of noise. The authors should explain why they chose the ones selected and not others, besides the well-recognized ones (Abeta and Tau)

Biomarkers: I think it is not worthy to include in the review biomarkers with little/controversial support (i.e: neurogranin…etc). As a matter of fact, I guess there a many other biomarkers that are not included for this same reason.

Table 2 is very useful.

While limitations of current animal models are discussed, a more in-depth discussion of their limitations and possible new avenues would be very convenient.

Minor: subheadings 2.9, 3.8 and 4.5 require more precise titles; or better, integrate them into the discussion.

Ref 162 refers to a transgenic rat model, not to a non-transgenic. Correct

Correct syntactic errors throughout the text

Author Response

Response to Reviewer 1 Comments

Point 1: In particular, the topic has an extensive literature, but the authors have favoured to include specific publications over many others. They should clarify why they chose to omit so many and the way they selected them (which is almost inevitable, given the enormous amount of data). They should also clearly state criteria to select specific biomarkers.

Response 1: we conducted the following search strategy for making the heat map and the biomarker part: Search in PubMed, Keywords: combinations of Alzheimer +human/animal model type + plasma OR blood OR CSF + biomarker type. Next we looked in literature found for other literature we might have missed. Up or down regulation was only included when it was statistically proven. Otherwise, as only a slight trend was seen, data was categorized as no change.

Point 2: Hypotheses of AD: should be nuanced. Thus, dysfunctional microglia is not hypothesized as the origin of AD; maybe pathogenic in the cascade, maybe not. This is still not clear.

Response 2: Actually we did not state that dysfunctional microglia is the origin of AD

Point 3: Oxidative stress may be contributing to the pathological cascade. It is no longer considered a mainstream cause of the disease.

Response 3: While it might be true that inflammation and oxidative stress are not the main cause of AD (which we did not imply, but the text might be read like that; we adapted), these processes seem to play an important role in the disease cascade.

Point 4: There are other hypotheses that are not mentioned, such as disturbed insulin function, dysregulated circuitry ….etc. If the authors considered them not worthy to be included, they should specify their selection criteria.

Response 4: We agree with the reviewer that there may be more processes underlying AD. We have adapted accordingly (Manuscript section 2.7)

Point 5: Table 1: Useful and interesting, but re-arrange it, more precision is needed. Many of the putative biomarkers are 1) arbitrarily assigned to a given hypothesis, or 2) arbitrarily added. I will elaborate: 1) For instance, Neurofilament, included in the “cholinergic” (outdated, by the way) hypothesis. Neurofilament is been considered as a marker of neuronal degeneration not only in AD but in neurodegenerative diseases in general (as commented by the authors also).

Response 5: The reviewer is right. It appeared to be remnant of a previous version with a different structure in the text etc. After updating the text, we overlooked the partly adapted table. This has now been corrected. So, thanks to the reviewer for this remark.

Point 6: Those added in the “Oxidative” hypothesis (calcineurin,CREB…) could be equally well included in other hypotheses, as they are involved in many biological pathways. 2) There are many more potential AD biomarkers reported. Maybe too many, making a lot of noise. The authors should explain why they chose the ones selected and not others, besides the well-recognized ones (Abeta and Tau)

Response 6: We agree, and adapted accordingly, please see previous response

Point 7: Biomarkers: I think it is not worthy to include in the review biomarkers with little/controversial support (i.e: neurogranin…etc). As a matter of fact, I guess there a many other biomarkers that are not included for this same reason.

Response 7: Actually, here we disagree with the reviewer. We included the most frequently studied biomarkers, as was indicated in the text. Therewith we indicated that it is not an excerpt of all the literature and studies on biomarkers in AD. We have now added that to the subheading to put more emphasis on that.

Point 8: Table 2 is very useful.

While limitations of current animal models are discussed, a more in-depth discussion of their limitations and possible new avenues would be very convenient.

Response 8: The limitations of using animal models have been addressed in the discussion, which we feel is the right place to do so.

Point 9 :subheadings 2.9, 3.8 and 4.5 require more precise titles; or better, integrate them into the discussion.

Response 9: Here we do not agree with the reviewer. It is the end part of a whole section, and we feel that a “taken together” is very important for the readability, so we have not changed.

Point 10: Ref 162 refers to a transgenic rat model, not to a non-transgenic. Correct

Response 10: Actually, we did state “transgenic”, so it was already correct.

Point 11: Correct syntactic errors throughout the text

Response 11: The manuscript has been revised on typos, text and readability

Reviewer 2 Report

In this review, Qin et al. attempt to make a case for animal studies of biofluid biomarkers in AD as an addition to the human studies currently being performed in humans.  They review different technologies being used for biomarker discovery, and how they may be combined to identify new biomarkers for AD.

In full disclosure, this is not a view I share, and this review does not successful persuade me that I should change my mind.  As the authors themselves admit, the mouse models of AD are built off familial cases, and share very few features of the more common sporadic late onset AD.  The pattern of amyloid and tau disposition is not reflective of human cases, and a large number of mutations are required in a mouse to result in gross neurodegeneration.  Furthermore, human AD has a huge comorbidity with other diseases such as heart disease and diabetes, and is the reflection of a lifetime of risk factors impossible to model well in a rodent.  All of this is covered at the end of the manuscript, and thus a case for prioritizing animal biomarkers is not made.  The only use I can see if for modelling very specific changes in biofluids – if we modify the ApoE4 allele for example, how does the biofluid profile change?  This is using the reductive nature of mouse models to its full strength – what can they tell us about the specific effect of genetic changes on the biofluid profile.

Major points

  • The major point of this review is supposed to be a comparison of human versus animal biofluid studies in AD, but presumably due to the reasons above and the small volumes of material available from animal studies, there are very few. So this is mostly a list of studies performed in human biofluids, tissue from both organisms, with very little intellectual input into these contrasts.  Furthermore, they state that AB42/AB40 ratio is the most critical measurement for diagnosis in the human population, but that the small number of animals studied do not show the same trends in this ratio.  The ratio is therefore not shown in the summary diagram Figure 1, because this finding puts the utility of this approach entirely into question. 

  • While I cannot speak to the completeness of the reviews on miRNAs and EVs, the proteomics section cherry picks a handful of low powered studies and ignores the top, well powered studies in this field such as Spellman et al.

  • All of the new technologies mentioned in the discovery section have significant drawbacks for use as robust biomarker quantification tools. Some discussion of their utility for discovery vs robust quantification would be welcomes. The EV techniques are multi stage processes (with huge questions about their validity – I am yet to hear of any groups outside the Goetzl sphere who can replicate these techniques, and the L1CAM work that was done in the early stages is not specific to neuronal exosomes, the early papers are missing all controls that would increase confidence in their work) with questions about how best to normalize between individuals and across time, and proteomics and metabolomics in discovery mode also share these issues. 

  • The splitting of the biomarkers by AD hypothesis is an interesting idea but proves artificial – SNAP-25, NfL and Neurogranin are not specific to cholinergic neurons, MAPKs and CAMKIIs are not directly reflective of only oxidative stress – there are many processes concomitantly occurring in AD that might change levels of all these proteins.

  • pTau is not a monolithic singular protein – it has many phosphorylation sites and multiple fragments in biofluids. Measurement is therefore highly dependent on the antibody pairs used, and the discussion of Tau is therefore hugely oversimplified.

Minor points

  • Page 2 line 48 should be MMP9, not MPP-9

  • The discussion of AB42/40 ratio in familial AD is not reflective of the reality – there are more studies than references and most show a decrease in the ratio with disease onset in CSF, with less agreement on the direction of change in plasma/serum. See Rosen Mol. Degeneration 2013.

  • The ATN framework is only for research purposes, it is not used for clinical patient stratification. The first sentences in this section sound like they were lifted directly from the paper introducing the framework.

  • How is metabolomics an essential biomarker technique if none of the initial findings using it have been validated?

Author Response

Response to Reviewer 2 Comments

Point 1: In full disclosure, this is not a view I share, and this review does not successful persuade me that I should change my mind. 

1.1 As the authors themselves admit, the mouse models of AD are built off familial cases, and share very few features of the more common sporadic late onset AD.  The pattern of amyloid and tau disposition is not reflective of human cases, and a large number of mutations are required in a mouse to result in gross neurodegeneration.

1.2 Furthermore, human AD has a huge comorbidity with other diseases such as heart disease and diabetes, and is the reflection of a lifetime of risk factors impossible to model well in a rodent.  All of this is covered at the end of the manuscript, and thus a case for prioritizing animal biomarkers is not made.

1.3 The only use I can see if for modelling very specific changes in biofluids – if we modify the ApoE4 allele for example, how does the biofluid profile change? This is using the reductive nature of mouse models to its full strength – what can they tell us about the specific effect of genetic changes on the biofluid profile.

Response 1:

1.1 We believe that the mouse model with most mutations has 3 mutations: 3xTg AD mice model (APP Swedish, MAPT P301L, and PSEN1 M146V see https://www.alzforum.org/research-models/3xtg). We did not look at the pattern of tau and amyloid deposition but rather to the trend: is there up/down regulation over time. We indeed state that the majority of animal models is transgenic and refer to familial cases, while most AD patients have the sporadic form. This is simple because transgenic animals are used so much, and therefore more information is present. But, also non-transgenic animal models have been constructed, that might relate to sporadic AD.  Furthermore, in human research we observe the same: the desperate clamping on the measurement of amyloid comes mainly of the knowledge of familial AD, as there it is known a mutation in the APP gene leads to a greater deposition of amyloid. This might play a role in sporadic AD but might not be that important in causality as is for familial AD.

1.2 Yes this is indeed true. But that is exactly why we need animal models: to study more than one piece of the puzzle all together and then look for causal relationships, in a way that cannot be readily studies in human AD. But once we have such information from animals, it may guide use to specific tests that can be done in humans, in which also knowledge on comorbidities can be taken into account.

1.3 Yes, we could specifically change the “condition” of an animal (by genetic modification, as you refer to, but you may also think of food, as well as of combined changes (See for example our study on ApoE (-/-) mice which, only when given also high fat diet, have alterations in blood-brain barrier functionality: Mulder et al. ApoE protects against neuropathol high fat diet BBB ageing. Lab Invest. 2001). So, indeed, it is of interest to study concomitant changes in biofluids due to changes in conditions (and combinations thereof): so integrated information. However, we typically see fragmentation information being published, and in our view that is what we should leave behind.  

Point 2: Furthermore, they state that AB42/AB40 ratio is the most critical measurement for diagnosis in the human population, but that the small number of animals studied do not show the same trends in this ratio.  The ratio is therefore not shown in the summary diagram Figure 1, because this finding puts the utility of this approach entirely into question.

Response 2: This is indeed the case. Actually we already provided specific data in the Supplement. Now it has also been specifically addressed in the text here (section 3.1 in manuscript).

Point 3: While I cannot speak to the completeness of the reviews on miRNAs and EVs, the proteomics section cherry picks a handful of low powered studies and ignores the top, well powered studies in this field such as Spellman et al.

Response 3: We definitely did not cherry pick proteomic studies. But, anyway, this section could be and have been extended accordingly (section 4.2 in manuscript).

Point 4: All of the new technologies mentioned in the discovery section have significant drawbacks for use as robust biomarker quantification tools. Some discussion of their utility for discovery vs robust quantification would be welcomes. The EV techniques are multi stage processes (with huge questions about their validity – I am yet to hear of any groups outside the Goetzl sphere who can replicate these techniques, and the L1CAM work that was done in the early stages is not specific to neuronal exosomes, the early papers are missing all controls that would increase confidence in their work) with questions about how best to normalize between individuals and across time, and proteomics and metabolomics in discovery mode also share these issues.

Response 4: We fully agree with the reviewer (for that reason, we are actually currently investigating all these issues). We have adapted the text to have it “spelled out”. Thanks for the remark (section 4.4 in manuscript). Also, in the text, we already provided these drawbacks. But, to have more emphasis on this we have added more (section 4.5 in manuscript).

Point 5: The splitting of the biomarkers by AD hypothesis is an interesting idea but proves artificial – SNAP-25, NfL and Neurogranin are not specific to cholinergic neurons, MAPKs and CAMKIIs are not directly reflective of only oxidative stress – there are many processes concomitantly occurring in AD that might change levels of all these proteins.

Response 5: For SNAP-25, NfL and Neurogranin, we agree with the reviewer, It appeared to be remnant of a previous version with a different structure in the text etc. After updating the text, we overlooked the partly adapted table. This has now been corrected. So, thanks to the reviewer for this remark. For MAPKs and CAMKIIs, the reviewer is fully right. We should have indicated this with a little more “degrees of freedom”. This actually is our vision (like we stated) that many processes are interlinked. This is exactly what we try to convey as a message, and why we propose to perform research in an integrated manner (see discussion)

Point 6: pTau is not a monolithic singular protein – it has many phosphorylation sites and multiple fragments in biofluids. Measurement is therefore highly dependent on the antibody pairs used, and the discussion of Tau is therefore hugely oversimplified.

Response 6: This is indeed the case. While we already provided specific p-tau information in the Supplement, we now also adapted the text to emphasize this point (section 3.2 in manuscript).

Point 7: Page 2 line 48 should be MMP9, not MPP-9

Response 7: This is now corrected.

Point 8: The discussion of AB42/40 ratio in familial AD is not reflective of the reality – there are more studies than references and most show a decrease in the ratio with disease onset in CSF, with less agreement on the direction of change in plasma/serum. See Rosen Mol. Degeneration 2013

Response 8: From the literature we took the ratio as calculated on the basis of cohorts of AD patients and sometimes MCI patients. We do not believe it is correct to compare literature on sporadic AD patients with that on patients with the mutation for familial AD but without AD phenotype. We can only mention that using the ratio provides extra information when comparing AD patients with healthy controls, but that the ratio cannot be used in familial AD subjects with the mutation but without complaints.

Point 9: The ATN framework is only for research purposes, it is not used for clinical patient stratification. The first sentences in this section sound like they were lifted directly from the paper introducing the framework.

Response 9: The reviewer is fully right. This should have been explicitly mentioned. We updated this in the text (section 2.8 in manuscript).

Point 10: How is metabolomics an essential biomarker technique if none of the initial findings using it have been validated?

Response 10: This had already been addressed implicitly (by ref on Hurtado, from our group), but should have been addressed explicitly, as has been done now (section 4.3 in manuscript).

Reviewer 3 Report

The manuscript proposed by de Lange and co-workers is well written and organized. It provides an overview on the studies based on the detection of biomarkers for AD in animal and human. It offers different insights for the investigation of relationship among biomarkers in brain and body fluids at different levels. I suggest its publication on the International Journal of Molecular Sciences after an in-depth revision of the text in order to correct some typos.

Author Response

Response to Reviewer 3 Comments

Point 1: The manuscript proposed by de Lange and co-workers is well written and organized. It provides an overview on the studies based on the detection of biomarkers for AD in animal and human. It offers different insights for the investigation of relationship among biomarkers in brain and body fluids at different levels. I suggest its publication on the International Journal of Molecular Sciences after an in-depth revision of the text in order to correct some typos.

Response 1: The manuscript has been revised on typos, text and readability

Round 2

Reviewer 2 Report

Review Summary - all reviewer comments in Italics: I do not believe that the authors have made a substantial response to my major criticism – that their argument is unpersuasive.  They have responded to my review by adding list-like sections of other studies, but have not addressed my major comment on their text – that this whole review proves that in fact animals are not adequate models of Alzheimer’s Disease, except in extremely reductive cases.  See specific comments embedded below.  There is a case to be made for the use of animal models in AD – but it is not made in this review.  It should be shorter, and punchier, and focus on the work such as this group have performed themselves – how can you actually leverage this reductive nature to highlight biomarkers that might be of great use to human studies?  Where are the knowledge gaps and how can animal studies help fill them?   There is an argument to be found here but it is not currently being well described, and proposing that this is a review rather than an opinion piece is actually somewhat misleading.

Response to Reviewer 2 Comments

Point 1: In full disclosure, this is not a view I share, and this review does not successful persuade me that I should change my mind.

1.1 As the authors themselves admit, the mouse models of AD are built off familial cases, and share very few features of the more common sporadic late onset AD. The pattern of amyloid and tau disposition is not reflective of human cases, and a large number of mutations are required in a mouse to result in gross neurodegeneration.

1.2 Furthermore, human AD has a huge comorbidity with other diseases such as heart disease and diabetes, and is the reflection of a lifetime of risk factors impossible to model well in a rodent. All of this is covered at the end of the manuscript, and thus a case for prioritizing animal biomarkers is not made.

1.3 The only use I can see if for modelling very specific changes in biofluids – if we modify the ApoE4 allele for example, how does the biofluid profile change? This is using the reductive nature of mouse models to its full strength – what can they tell us about the specific effect of genetic changes on the biofluid profile.

Response 1:

  • We believe that the mouse model with most mutations has 3 mutations: 3xTg AD mice model (APP Swedish, MAPT P301L, and PSEN1 M146V see https://www.alzforum.org/research-models/3xtg). We did not look at the pattern of tau and amyloid deposition but rather to the trend: is there up/down regulation over time. We indeed state that the majority of animal models is transgenic and refer to familial cases, while most AD patients have the sporadic form. This is simple because transgenic animals are used so much, and therefore more information is present. But, also non-transgenic animal models have been constructed, that might relate to sporadic AD. Furthermore, in human research we observe the same: the desperate clamping on the measurement of amyloid comes mainly of the knowledge of familial AD, as there it is known a mutation in the APP gene leads to a greater deposition of amyloid. This might play a role in sporadic AD but might not be that important in causality as is for familial AD.

I’m not actually sure what most of this rebuttal paragraph means.  It is not true that most human work has been focused on genetic forms, it is true that for too long amyloid has been the sole focus of new therapeutics.  But even in familial forms of AD pathology progresses through specific regions in a way that mouse models simply don’t reflect.  This rebuttal doesn’t persuaded me to look more at animal models, it persuades me that we need to do bigger, more robust studies of humans with AD.

  • Yes this is indeed true. But that is exactly why we need animal models: to study more than one piece of the puzzle all together and then look for causal relationships, in a way that cannot be readily studies in human AD. But once we have such information from animals, it may guide use to specific tests that can be done in humans, in which also knowledge on comorbidities can be taken into account.

But this is not the case being made in the review – the review is simply comparing them and falling short. 

  • Yes, we could specifically change the “condition” of an animal (by genetic modification, as you refer to, but you may also think of food, as well as of combined changes (See for example our study on ApoE (-/-) mice which, only when given also high fat diet, have alterations in blood-brain barrier functionality: Mulder et al. ApoE protects against neuropathol high fat diet BBB ageing. Lab Invest. 2001). So, indeed, it is of interest to study concomitant changes in biofluids due to changes in conditions (and combinations thereof): so integrated information. However, we typically see fragmentation information being published, and in our view that is what we should leave behind.

Again, the review does not talk about this.  This is a far stronger argument than the one proposed in the review.  The only time this approach is mentioned is towards the very end, when this is stated “Although an ideal animal model exhibiting all features of human AD does not exist, and 737 current animal AD models are at best to be regarded as reductionist tools to gain understanding on 738 the pathological processes involved in AD, we believe that animal studies hold important 739 advantages and potential.”   The purpose of the review is buried right at the end, and the rest of the review does not build to this argument.  This is what I meant by major revisions – not an additional of more lists of biomarkers – other people have done this work already.

Point 2: Furthermore, they state that AB42/AB40 ratio is the most critical measurement for diagnosis in the human population, but that the small number of animals studied do not show the same trends in this ratio. The ratio is therefore not shown in the summary diagram Figure 1, because this finding puts the utility of this approach entirely into question.

Response 2: This is indeed the case. Actually we already provided specific data in the Supplement. Now it has also been specifically addressed in the text here (section 3.1 in manuscript).

Okay – but how does this change your argument?

Point 3: While I cannot speak to the completeness of the reviews on miRNAs and EVs, the proteomics section cherry picks a handful of low powered studies and ignores the top, well powered studies in this field such as Spellman et al.

Response 3: We definitely did not cherry pick proteomic studies. But, anyway, this section could be and have been extended accordingly (section 4.2 in manuscript).

This is not an academic review of this literature – it is a list and does not build on the central argument of the review.  Others have done a more complete review of this literature – it is fine to reference them.  My argument would be that there is not much agreement between this literature, and that mouse models could be used to test the relevance of those targets that are coming out of multiple studies.

Point 4: All of the new technologies mentioned in the discovery section have significant drawbacks for use as robust biomarker quantification tools. Some discussion of their utility for discovery vs robust quantification would be welcomes. The EV techniques are multi stage processes (with huge questions about their validity – I am yet to hear of any groups outside the Goetzl sphere who can replicate these techniques, and the L1CAM work that was done in the early stages is not specific to neuronal exosomes, the early papers are missing all controls that would increase confidence in their work) with questions about how best to normalize between individuals and across time, and proteomics and metabolomics in discovery mode also share these issues.

Response 4: We fully agree with the reviewer (for that reason, we are actually currently investigating all these issues). We have adapted the text to have it “spelled out”. Thanks for the remark (section 4.4 in manuscript). Also, in the text, we already provided these drawbacks. But, to have more emphasis on this we have added more (section 4.5 in manuscript).

Okay for EVs – what about the other techniques such as proteomics?  If  you do not have expertise in these methods then there are reviews out there that spell this out – eg, Carlyle, 2018 Proteomes.  These can be referenced to build the central story that these discovery tools are not good for robust repeat measurements, and that mouse models can be used to refine the targets lists. 

Point 5: The splitting of the biomarkers by AD hypothesis is an interesting idea but proves artificial – SNAP-25, NfL and Neurogranin are not specific to cholinergic neurons, MAPKs and CAMKIIs are not directly reflective of only oxidative stress – there are many processes concomitantly occurring in AD that might change levels of all these proteins.

Response 5: For SNAP-25, NfL and Neurogranin, we agree with the reviewer, It appeared to be remnant of a previous version with a different structure in the text etc. After updating the text, we overlooked the partly adapted table. This has now been corrected. So, thanks to the reviewer for this remark. For MAPKs and CAMKIIs, the reviewer is fully right. We should have indicated this with a little more “degrees of freedom”. This actually is our vision (like we stated) that many processes are interlinked. This is exactly what we try to convey as a message, and why we propose to perform research in an integrated manner (see discussion)

There is still the need for evidence to link general markers of neuron/brain function to specific hypotheses if these links already exist.  If not, then it needs to be made abundantly clear that this is their proposed framework, and that animal models can be used to test these hypotheses.  This is not currently abundantly clear.

Point 6: pTau is not a monolithic singular protein – it has many phosphorylation sites and multiple fragments in biofluids. Measurement is therefore highly dependent on the antibody pairs used, and the discussion of Tau is therefore hugely oversimplified.

Response 6: This is indeed the case. While we already provided specific p-tau information in the Supplement, we now also adapted the text to emphasize this point (section 3.2 in manuscript).

Okay.

Point 8: The discussion of AB42/40 ratio in familial AD is not reflective of the reality – there are more studies than references and most show a decrease in the ratio with disease onset in CSF, with less agreement on the direction of change in plasma/serum. See Rosen Mol. Degeneration 2013

Response 8: From the literature we took the ratio as calculated on the basis of cohorts of AD patients and sometimes MCI patients. We do not believe it is correct to compare literature on sporadic AD patients with that on patients with the mutation for familial AD but without AD phenotype. We can only mention that using the ratio provides extra information when comparing AD patients with healthy controls, but that the ratio cannot be used in familial AD subjects with the mutation but without complaints.

The paper I mentioned previously does review more studies of familial AD and it contradicts what you have written. 

Point 10: How is metabolomics an essential biomarker technique if none of the initial findings using it have been validated?

Response 10: This had already been addressed implicitly (by ref on Hurtado, from our group), but should have been addressed explicitly, as has been done now (section 4.3 in manuscript).

This is a good addition.

Round 3

Reviewer 2 Report

I believe we are at a point of fundamental misunderstanding about this review.  In the rebuttal letter, the authors state: "Then with regard on writing a review that, according to the reviewer, should focus more on our work. In all the years that I have been performing research on a topic it always has started with a review on the problem and how we envisioned what should be done to help solving the problem, while in the meantime we were working on that. So the theses of our PhD students typically start with a review that sets the stage, and then followed by research articles. It is a great way of having the PhD students defining there nice in contribution to solve the problem in a more focused manner. I would say that that is good scientific practice and this is what we do. I do not see any problem here."

Of course, this is an important thing for any PhD student to do before they begin a research project on a theme.  However, this does not mean that a published version of the review will contribute substantially to the body of academic literature.  My consistent criticism is that this review does not do what it claims to - provide a persuasive case for the use of animals in defining biomarkers for Alzheimer's Disease, particularly in the early stages.  It is getting longer and more unmanageable, yet fundamentally fails to address my criticism.  In what specific cases have animal models taught us about Alzheimer's Disease biomarkers?  How have they helped in other complex diseases that we can learn from Alzheimer's Disease?  This review basically says that people have only looked at the core AD biomarkers in both humans and animals, they only agree sometimes, and so we should do more animal work.  That is fundamentally not a persuasive argument.  

The argument in the rebuttal that it is too expensive to do large human studies is also not true.  The ROSMAP, Massachusetts ADRC, Harvard Aging Brain Project, Lothian Birth Cohort, Honolulu-Asia, BLSA, DIAN, Emory brainbank, and numerous other collections are all running longitudinal cohorts that have proved absolutely critical to our understanding of Alzheimer's Disease, and how biomarkers perform in AD. Animal studies have yet to add to this picture.  

Regarding the table that places biomarkers into theories of AD - my criticism is not that the biomarkers mentioned are not specific to AD, it is that they are not specific to the pathways that the table connects them to.  Calcineurin, CREB, and MAPK are not solely related to oxidative stress, they are vital for normal function of a neuron, for signaling, for LTP, for calcium regulation - to posit them as being solely connected to oxidative stress is incorrect.  

I do not believe I can offer any more advice as the authors are not listening to my constructive criticisms.  While this review may be acceptable in its original form as an introduction to a thesis, it does not contribute to the body of academic literature on this subject.  

As a final note, in writing these rebuttals, the authors should take care avoid using gendered terms - I recommend the use of the word "their" in place of "his": "Overall, we really want to thank reviewer 2 for his critical view and comments that have been very important for improving the review. " 

Author Response

Point-to-point response to Reviewer2’s comments in blue

I believe we are at a point of fundamental misunderstanding about this review.  In the rebuttal letter, the authors state: "Then with regard on writing a review that, according to the reviewer, should focus more on our work. In all the years that I have been performing research on a topic it always has started with a review on the problem and how we envisioned what should be done to help solving the problem, while in the meantime we were working on that. So the theses of our PhD students typically start with a review that sets the stage, and then followed by research articles. It is a great way of having the PhD students defining there nice in contribution to solve the problem in a more focused manner. I would say that that is good scientific practice and this is what we do. I do not see any problem here."

Of course, this is an important thing for any PhD student to do before they begin a research project on a theme.  However, this does not mean that a published version of the review will contribute substantially to the body of academic literature.  My consistent criticism is that this review does not do what it claims to - provide a persuasive case for the use of animals in defining biomarkers for Alzheimer's Disease, particularly in the early stages.  It is getting longer and more unmanageable, yet fundamentally fails to address my criticism.  In what specific cases have animal models taught us about Alzheimer's Disease biomarkers?  How have they helped in other complex diseases that we can learn from Alzheimer's Disease?  This review basically says that people have only looked at the core AD biomarkers in both humans and animals, they only agree sometimes, and so we should do more animal work.  That is fundamentally not a persuasive argument. 

Responses (for parts of the text above):

“Of course, this is an important thing for any PhD student to do before they begin a research project on a theme.  However, this does not mean that a published version of the review will contribute substantially to the body of academic literature”.

Response: We have enough evidence where our reviews (with the same nature like this) one have been welcomed and really contributed to the academic discussion. We take science very seriously.

In this particular case, 3.5 years ago, the PhD student started on a project intended for mechanistically model AD disease progression, using data to be extracted from public sources (publications, databases), with an emphasis on use of EV based information. We came across many inconsistencies in literature, also with regard to methodologies used, with how information was quantified, etc, such that the value of the data was far too low for being able to develop a good AD progression model.

That was a huge problem and we decided to shift gears:  to actually produce data ourselves in a highly systematic and structured manner, in well-defined conditions, using the same methodologies for comparing data, etc (The mastermind Research Approach), which is a need and that is the message that we really feel should get across.

And currently we are working accordingly, however not yet ready for being submitted for publication.  But, in developing the generally applicable CNS drug distribution model) as indicated as an approach in the review, despite the initial lack of understanding of colleagues in the field, the use of this approach has been proven a real success.

“My consistent criticism is that this review does not do what it claims to - provide a persuasive case for the use of animals in defining biomarkers for Alzheimer's Disease, particularly in the early stages”.

Response: We do not claim that animals should be used for defining biomarkers for AD, especially in early stage. We wrote the following (see abstract):

  • To diagnose and to treat early stage (preclinical) AD patients, we need body fluid based biomarkers that reflect the processes that occur in this stage, but current knowledge on associated processes is lacking. As human studies on (possible) onset and early stage AD would be extremely expensive and time consuming, we investigate the potential value of animal AD models to help to fill this knowledge gap
  • This indicates the potential value of animal AD models in understanding of also onset and early AD. Moreover, animal studies can be smartly designed to provide mechanistic information on the interrelationships between the different AD processes in a longitudinal fashion, and may also include the combinations of different conditions that may reflect comorbidities in human AD; according to the Mastermind Research approach.

And in the discussion:

  • (line 491). For early diagnosis and treatment of AD, we first need to solve the problem of the gap of knowledge and understanding of the onset and early phase of AD.
  • (line 510). We believe that the problem of the gap of knowledge and understanding of the onset and early phase of AD cannot be solved by human studies alone. This is for the simple reasons that it is too costly and too time-intensive to measure many compounds in human samples, let alone imaging studies, in well-controlled and longitudinal studies in the ageing human population, in which a small percentage will actually develop AD
  • (line 569). Altogether, in our view, strategic and well-controlled animal studies are needed to fill the crucial knowledge gap especially on the processes involved in the onset and early AD. Thus, as much as possible in individual animals, multiple (putative) biomarkers should be measured at multiple body sites including body fluids, to understand their interdependencies and time-dependencies in AD onset and progression (according to the Mastermind Research Approach). By this approach also systematic research can be performed on combination of conditions, such as comorbidities.

“It is getting longer and more unmanageable….. ”

Response: on the contrary; the length of the manuscript has been substantially reduced. It might be that the version with track changes gave the impression that it was getting larger, but it really did not.

“…  yet fundamentally fails to address my criticism.  In what specific cases have animal models taught us about Alzheimer's Disease biomarkers?  How have they helped in other complex diseases that we can learn from Alzheimer's Disease?  This review basically says that people have only looked at the core AD biomarkers in both humans and animals, they only agree sometimes, and so we should do more animal work.  That is fundamentally not a persuasive argument.

Response: The fact that after so many investments and research in animals and human of the last decade without any successful treatment and no early diagnosis, indicates that things should be done differently. We feel AD as a real problem and we have thought about gaps in knowledge and how to fill those without having to wait for the long time for disease progression in human, and then trace back, and learn from that in a slow way, while also costs of having humans that might develop AD to be incorporated in studies, that will lack individual linkage of processes happening in brain to that what can be seen in body-fluids. The problem is large enough to look for additional possibilities, opportunities, contributions…  so we did.

On the basis of the data so far found in literature we conclude that, where comparison is more or less possible, animals display a same trend in changes (or no changes) in biomarkers. So there seems to be value in animal studies. However, not just as such. We plea for a better use of animals, by performing studies that will provide multiple-level connected data for understanding time-dependencies and interrelationships of processes contributing to AD? Such integrated information can be obtained from animals in cheaper and faster way than in humans. Also, in our review we look to the future possibilities, and have therefore included emerging biomarkers in AD. Interestingly, these emerging biomarkers have been studies more in animals than in humans, as compared to the “traditional biomarkers”. So, with better designed animal studies, we can learn from the animal models as much as possible, and translate the findings to human diagnosis of AD.

In order to make ourselves clearer, we have adapted line 132 and further into:

“So, the problem is how to get more mechanistic information on the time course and interrelationships on the rate and extent of processes that drive the onset and early development of human AD. In humans, there is the possibility for monitoring blood levels of multiple body compounds (potential biomarkers) in cohort. Many of such cohort measurements are currently ongoing. Although we might learn a lot from such studies, there are crucial limitations. First, for detecting early changes in body processes that may lead to AD, plasma information is not sufficient, as the levels of body compounds may result to many disturbances not necessarily connected to AD onset. Information on the brain might be provided by what can be detected using imaging techniques. However, imaging techniques are very costly and will not be used in all subjects, let alone in each human subject at each year of follow-up. Then, human subjects should have developed (significant indicators of) AD, before the information of the human subject in relation to AD progression can be made. As AD progresses slowly, this will take at least years. In other words, such studies at best would be very expensive and time-consuming. So, what could be done in addition, as alternative approaches to help solve the problem?”

The argument in the rebuttal that it is too expensive to do large human studies is also not true.  The ROSMAP, Massachusetts ADRC, Harvard Aging Brain Project, Lothian Birth Cohort, Honolulu-Asia, BLSA, DIAN, Emory brain bank, and numerous other collections are all running longitudinal cohorts that have proved absolutely critical to our understanding of Alzheimer's Disease, and how biomarkers perform in AD. Animal studies have yet to add to this picture. 

Response: We did not state that it is too expensive to do large human studies. We are aware of the large human studies. We indicated time and cost limitations of human studies in producing knowledge that will help to understand the time-dependencies and interrelationships of (in particular early) processes contributing to AD. Unfortunately, biomarker performance as such is not enough. And human brain bank tissues are postmortem human brain tissues that typically will not reflect early AD processes, while we can obtain brain tissue form early AD animal models. Therefore, we feel we should be openminded to additional possibilities to tackle the limitations of human studies, by performing (high quality designed ) animal studies.

Regarding the table that places biomarkers into theories of AD - my criticism is not that the biomarkers mentioned are not specific to AD, it is that they are not specific to the pathways that the table connects them to.  Calcineurin, CREB, and MAPK are not solely related to oxidative stress, they are vital for normal function of a neuron, for signaling, for LTP, for calcium regulation - to posit them as being solely connected to oxidative stress is incorrect. 

Response. There is no (or hardly any) body compound that is solely related to one pathway, while also the pathways as such are not exclusive to health or disease. Disease is the result of the imbalance in the rate and extent of body processes, beyond the “resilience boundaries”, and a change in the level of a body compound may reflect such imbalance (disease).  In Table 1, we want to link potential body fluid based biomarkers to disease process in AD, this requires the marker in plasma to be related to an imbalance in rate and extent of processes in the brain related to AD, with the potential to be detected  body fluids above the noise of interference from these processes in the periphery. Therefore, the general oxidative stress markers such as ROS and RNS are indicated to be not specific enough to represent the oxidative stress in CNS/AD.  

For the kinases we admit their function in normal conditions, but we are more interested in how these markers (pathways) change in oxidative stress response in AD. To balance the input from the reviewer but avoid to expand the discussion to introduce all the functions and relations of these markers, now we add the specific pathways that connecting these markers to oxidative stress and typical pathologies in AD (tau phosphorylation, Aβ deposition, synaptic dysfunction..ect.) in our Table 1.

I do not believe I can offer any more advice as the authors are not listening to my constructive criticisms.  While this review may be acceptable in its original form as an introduction to a thesis, it does not contribute to the body of academic literature on this subject. 

Response. The editor may check for all the criticism provided by the reviewer that have been addressed in the previous rounds. By that, a lot of improvements have been made and we honestly thank the reviewer for that (see also the last note).

As a final note, in writing these rebuttals, the authors should take care avoid using gendered terms - I recommend the use of the word "their" in place of "his": "Overall, we really want to thank reviewer 2 for his critical view and comments that have been very important for improving the review. "

Response: I all the rebuttal text we have used “the reviewer”, in a gender-neutral fashion, but you got us on this mistake. We know and feel the importance of that, and hope that the message as such, our sincere thanks for the critical view, is what will be remembered.